# Interactions between Avibactam and Ceftazidime-Hydrolyzing Class D β-Lactamases

**DOI:** 10.3390/biom10030483

**Published:** 2020-03-23

**Authors:** Jean-Marie Frère, Pierre Bogaerts, Te-Din Huang, Patrick Stefanic, Joël Moray, Fabrice Bouillenne, Alain Brans

**Affiliations:** 1Centre for Protein Engineering, University of Liège, B 4000 Liège, Belgium; pstefanic@uliege.be (P.S.); jmoray@uliege.be (J.M.); f.bouillenne@uliege.be (F.B.); abrans@uliege.be (A.B.); 2National Reference Laboratory for Monitoring of Antimicrobial Resistance in Gram-Negative Bacteria, CHU Dinant-Godinne, UCL Namur, B 5530 Yvoir, Belgium; pierre.bogaerts@uclouvain.be (P.B.); te-din.huang@uclouvain.be (T.-D.H)

**Keywords:** class D β-lactamases, OXA-β-lactamases, OXA-24, OXA-163, OXA-427, avibactam, ceftazidime

## Abstract

Class D β-lactamases exhibit very heterogeneous hydrolysis activity spectra against the various types of clinically useful β-lactams. Similarly, and according to the available data, their sensitivities to inactivation by avibactam can vary by a factor of more than 100. In this paper, we performed a detailed kinetic study of the interactions between two ceftazidime-hydrolyzing OXA enzymes and showed that they were significantly more susceptible to avibactam than several other class D enzymes that do not hydrolyze ceftazidime. From a clinical point of view, this result is rather interesting if one considers that avibactam is often administered in combination with ceftazidime.

## 1. Introduction

β-lactam antibiotics (mostly penicillin, cephalosporins, carbapenems and monobactams) account for almost two-thirds of the antimicrobial agents prescribed at the hospital level [1]. In Enterobacterales and non-fermenters (especially *Pseudomonas aeruginosa* and *Acinetobacter baumannii*), bacterial resistance to β-lactams is mainly due to the production of β-lactamases which inactivate β-lactams by hydrolysis of the β-lactam ring [1]. On the basis of their sequences, β-lactamases belong to one of four classes: class A (for example the mycobacterial β-lactamases, the TEM, SHV or CTX-M extended-spectrum β-lactamases (ESBLs) and the KPC-carbapenemases), class B (metallo-β-lactamases: the archetypal *Bacillus cereus* enzyme and the IMP, NDM, VIM carbapenemases), class C (chromosome-encoded enzymes, produced by many Gram-negative strains and plasmidic AmpC’s such as CMY-2-like and DHA) or class D (mainly OXA-β-lactamases) (for detailed reviews on β-lactamases, see [2,3] and several papers in the present volume). Classes A, C and D consist of active-site serine β-lactamases for which various inactivators (e.g., clavulanate, tazobactam and avibactam) have been described. These can be used in combination with β-lactams to inhibit β-lactamases and restore their antimicrobial activities [2,4]. However, the efficiency of these inactivators can also be extremely variable depending on the particular target enzyme. For instance, clavulanic acid which is generally active against class A β-lactamases, is a very poor inhibitor of the class A KPC carbapenemase and of enzymes belonging to classes C and D [2,4]. It is even a substrate of the class B metallo-β-lactamases [5]. Plasmid-borne β-lactamase genes that can spread rapidly by horizontal gene transfer represent the major threat. These can be found in all classes and many OXA-type enzymes are plasmid-mediated.

Class D β-lactamases constitute a large group of heterogeneous enzymes including variants able to hydrolyze carbapenems, the last resort antibiotics (CHDLs: Carbapenem hydrolyzing Class D β-lactamases). In particular, OXA-23-like, OXA-24-like or OXA-58-like enzymes are the most prevalent variants encountered in the non-fermenter *Acinetobacter baumannii*. In Enterobacterales, the major threat arises from the OXA-48-like family. Interestingly, in contrast to the majority of carbapenemases of other classes such as the class A KPC variants or class B metallo-enzymes, CHDL enzymes present only a very low activity against expanded-spectrum cephalosorins such as ceftazidime. Intriguingly OXA-163, a variant of OXA-48 that presents a four-amino-acid deletion compared to OXA-48, has lost its carbapenemase activity [6,7,8] while expanding its hydrolysis spectrum to ceftazidime. OXA-163 is hence considered rather as an ESBL than as a carbapenemase. This last point is nevertheless still debated as the hydrolytic activity of OXA-163 seems sufficient to confer carbapenem resistance in specific genetic backgrounds (e.g., low outer membrane permeability [8,9,10]) and it is still considered as a carbapenemase by the Clinical and Laboratory Standards Institute (CLSI). OXA-163 is rarely reported in Western countries but is most prevalent in South America and North Africa (Egypt) where it represents an important clinical issue [11,12]. Another rarely reported but interesting CHDL variant is OXA-427 which presents both ESBL and carbapenemase activities [13,14,15]. This enzyme presents a 77% amino acid identity with OXA-12 from *Aeromonas sobria* but only 29% with OXA-48. The OXA-427 coding gene was possibly mobilized into an IncA/C plasmid from the chromosome of *Aeromonas media* WS, a strain used industrially in China to produce melanin. OXA-427 initially spread in a single Belgian clinical setting during a large outbreak that occurred in 2012 and is still occasionally detected in carbapenem-resistant Enterobacterales in Belgium (Bogaerts, unpublished data). Currently, OXA-163 and OXA-427 are quite scarce and more difficult to detect than many other CHDLs. For this reason, it is not excluded that they could spread silently and remain undetected until they largely emerge taking advantage of some environmental or genetic favorable combinations. A compound that can inactivate them would thus be quite useful and avibactam is one of the few CHDL inhibitors at our disposal to potentially fight these enzymes. However, avibactam does not appear to be an efficient general inactivator of OXA enzymes [16,17]. Ehmann et al. [16] have shown OXA-48 to be rather susceptible to inactivation by this compound (k_2_/K = 1400 M^−1^s^−1^) while OXA-10 was not (k_2_/K = 11 M^−1^s^−1^). Since avibactam is usually combined with ceftazidime, a study of the susceptibility to avibactam of two OXA ceftazidime hydrolyzing enzymes, OXA63 and Oxa-427 can represent a valuable approach together with a comparison to the interaction between avibactam and the OXA-24 carbapenemase for which a k_2_/K value of 52 M^−1^s^−1^ has been reported [17] but that was not found to exhibit very significant ceftazidime-hydrolyzing properties [18].

Unfortunately, there are very few other reliable published data about the hydrolysis of ceftazidime by OXA enzymes and about the interactions between these enzymes and avibactam. In the present study, we determine the kinetic parameters of the hydrolysis of ceftazidime by OXA-163 and OXA-427 and show that these two enzymes are reasonably well inactivated by avibactam, a result that might be of clinical significance.

## 2. Materials and Methods

### 2.1. Compounds

Avibactam was obtained successively from AstraZeneca (Waltham, MA, USA) and Pfizer (New York, NY, USA) and ceftazidime from AstraZeneca. Cephalothin was purchased from Sigma-Aldrich (Overijse, Belgium) and nitrocefin from Oxoid.

### 2.2. Enzymes

OXA-24 was produced as described by Bou et al. [18]. The stock solution was at 2.7 mg/mL (100 µM). All experiments were performed in a 90 mM sodium phosphate buffer, pH 7.0, containing 40 mM NaHCO_3_.

OXA-163: The gene was obtained from GeneCust Europa, L3505, Dudelange, Luxemburg. The sequence was based on the data of Poirel et al. [8] and that of the protein (confirmed by mass spectrometry, see the Results section) is shown in Figure A1. Note that a (His)_6_ tail was added to facilitate the purification. The molecular mass is 28,502.2 Da. The gene was introduced in pET28 between the NdeI and HindIII sites and the resultant plasmid introduced into *Escherichia coli* BL 21 (DE3) cells. For production, 9 × 25 mL of culture were grown in 100 mL Erlenmeyers (LB + kanamycin at 30 mg/L). Induction was performed with 1 mM IPTG at an A_600_ value of 1.0 and the culture continued for 4 h at 37 °C. Purification was performed on a Ni-Histrap HP column. The pure fractions were collected and concentrated to 1.12 mg/mL (39 µM) and passed through a 0.22 µM filter. The preparation was more than 95% pure on the basis of SDS-gel electrophoresis (not shown). All kinetic measurements were performed in 90 mM sodium phosphate, pH 7.0, containing 40 mM NaHCO_3_.

OXA-427: The original gene [13] was modified to yield a protein in which the signal peptide (residues 1–22) was replaced by a methionine residue for intracellular production and a VE(H)_8_ sequence added at the C-terminus to facilitate the purification (plasmid pET41-b-OXA-427). The sequence of the produced protein is shown in Figure A2. Note the loss of the N-terminal methionine that was confirmed by mass spectrometry (see the Results section). A 2-L overnight culture of *E. coli* BL21 harboring the plasmid was grown in BHI broth containing 50 mg/L kanamycin, the cells collected by centrifugation, resuspended in 50 mM Na phosphate buffer, pH 7.0, and disrupted by sonication.

After clarification of the solution by centrifugation, purification was performed on a 5 mL Ni-NTA column, followed by desalting with the help of a GO25 molecular sieve column in 50 mM sodium phosphate buffer, pH 7.0. The final enzyme concentration was 27 µM. The enzyme was at least 95% pure (SDS gel electrophoresis, not shown).

For the buffers used in the experiments, see the Results section.

### 2.3. Kinetic Measurements

All experiments were performed in triplicate at 30 °C. When the enzymes were diluted at concentrations below 0.1 mg/mL, bovine serum albumin was added at a concentration of 50 µg/mL. Hydrolysis of ceftazidime and cephalothin was monitored at 260 nm and that of nitrocefin at 485 nm.

The kcat and K_m_ values for nitrocefin, ceftazidime and cephalothin were determined under initial rate conditions or by analysis of the complete time-courses (as indicated in the Results section, see [19] for the analysis of complete time-courses).

The interaction with avibactam was studied on the basis of the model (Scheme 1) proposed by Ehmann et al. [16]:

In this model,
k_i_ = k_−2_ + k_f_(1)
where
k_f_ = k_2_[Avi]/(K+[Avi])(2)

Inactivation experiments (K, k_2_): the enzymes were incubated with various concentrations of avibactam in the presence of an adequate reporter substrate. The curves were analyzed with the help of the home-made Kinetics program based on the paper by De Meester et al. [19]. This program allows for the computation of the k_i_ values on the basis of the decrease in the rate of utilization of the reporter substrate. If substrate depletion results in a significant decrease in the rate in the control samples (no avibactam), this can be taken account of. For OXA-24, the reporter substrate was nitrocefin, for OXA-163, ceftazidime and for OXA-427, both ceftazidime and cephalothin. When the reporter substrate concentration is not much smaller than its K_m_ value, a correction must be introduced to account for protection by the substrate:(k_f_)_corr_ = (k_f_)_obs_ × (K_m_ + [S])/K_m_ and (k_2_/K)_corr_ = (k_2_/K)_obs_ × (K_m_ + [S])/K_m_(3)

Reactivation experiments (k_−2_) were performed by incubating the enzymes with avibactam concentrations sufficient to result in complete inactivation. Whenever possible, samples were then diluted 1000-fold or more in order to decrease the avibactam concentration below the global equilibrium constant and the recovered activity was measured on aliquots sampled after various periods of time. For OXA-427, the reactivation rate was difficult to determine because of the relatively poor activity of the enzyme (the k_cat_/K_m_ value of ceftazidime, that is one of the best substrates, is only 85,000 M^−1^s^−1^) and the large dilution factors necessary to significantly decrease the concentration of avibactam resulted in very low activities, poor stability of the protein and, in consequence, very large errors (see the Results section).

Global equilibrium constants (K_eq_ = k_−2_K/k_2_): whenever possible, the enzymes were incubated in the presence of avibactam concentrations somewhat higher than the expected K_eq_ value and the residual activity determined. The time of contact was calculated in order to reach at least 95% of the expected equilibrium value.

### 2.4. Mass Spectrometry

Experiments were performed on a Waters Easy-Q-ToF, SYNAPT G2 HDMS, in the positive ion mode. Samples, usually about 25 µM in protein, were first dialyzed against 50 mM NH_4_HCO_3_ and diluted 4-fold with a mixture of formic acid, acetonitrile and ammonium acetate to yield final concentrations of 0.5% formic acid and 30% or 50% acetonitrile. The expected mass accuracy is 100 ppm.

## 3. Results

### 3.1. OXA-24

#### 3.1.1. Interaction with Ceftazidime and Nitrocefin

Bou et al. [18] give no details on these substrates. The *E. coli* TG1 MIC for ceftazidime increases two-fold when the cloned enzyme is produced. This is not significant since only four-fold (2 dilutions) is considered as a significant MIC difference between 2 isolates. In the presence of 4 µM enzyme, and starting with 80 µM ceftazidime, the hydrolysis rate is below 0.002 µM/s. The K_m_ value was estimated as a K_i_ by incubating 1 nM enzyme with 40 µM nitrocefin and 0, 500 and 960 µM ceftazidime and monitoring the absorbance at 482 nm. Residual activities were 51% at 500 µM and 32% at 960 µM indicating a K_m_ value of 310 ± 20 µM. On this basis, it can be concluded that k_cat_ is < 2.5 × 10^−3^ s^−1^ and k_cat_/K_m_ < 8 M^−1^s^−1^. Clearly, ceftazidime cannot be used as a reporter substrate.

Preliminary experiments indicated that nitrocefin was a good substrate. From complete time-courses determined with 70 µM nitrocefin, the K_m_ value was estimated at 63 ± 28 µM with k_cat_ = 550 ± 230 s^−1^.

#### 3.1.2. Inactivation by Avibactam in the Presence of 70 µM Nitrocefin

The enzyme concentration was 1 nM. The concentrations of avibactam ranged from 133 to 400 µM. The observed inactivation rate constants varied from 0.0038 to 0.0109 s^−1^ and were proportional to the avibactam concentration since extrapolation of the line at [Avi] = 0 was very close to zero so that Equations (1) and (2) simplified to:k_i_ = k_f_ = (k_2_/K)_obs_[Avi](4)

This yielded a (k_2_/K)_obs_ value of 26 ± 3 M^−1^s^−1^ corresponding to a (k_2_/K)_corr_ value of 55 ± 5 M^−1^s^−1^ after correction for protection by the reporter substrate (Equation (3)). Moreover, k_−2_ is significantly lower than 0.001 s^−1^. This is in excellent agreement with the results of Lahiri et al. [17]. These results also show that K is significantly larger than 400 µM.

#### 3.1.3. Reactivation Rate

A preliminary experiment indicated that activity recovery was very slow and that the half-life of the adduct was likely to be >24 h. The enzyme (1 µM, 100 µL) was incubated with 9 µM avibactam. After 90 min, the solution was diluted 450-fold (residual avibactam concentration: 20 nM) and the activity determined on 450 µL aliquots by the addition of 50 µL of 400 µM nitrocefin. A control sample without avibactam was similarly treated but diluted 1575-fold. Activity was determined after 0, 21 (recovery: 22%) and 45 (recovery: 36%) h. To calculate the k_−2_ value, it must be considered that the measured rate constant = k_−2_ + k_f_ and that the recovery is limited by the equilibrium constant:% Activity at equilibrium = 100/(1 + [Avi]/K_eq_)(5)

Successive approximations indicate that K_eq_ is close to 60 nM. On this basis, the activity at equilibrium is expected to be 74% of that of the control (note that we did not observe any decrease of the activity of the control over the 45 h period) and the recovery rate constant value is 4.2 × 10^−6^ s^−1^. The k_f_ value at 20 nM avibactam is 1 × 10^−6^ s^−1^ so that k_−2_ is 3.2 × 10^−6^ s^−1^ (half-life = 60 h). The global equilibrium constant (k_−2_K/k_2_) is thus 3.2 × 10^−6^ M/55 = 58 nM.

#### 3.1.4. Global Equilibrium Constant

A value of 58 nM is given above, but due to the very low value of k_−2_, it might be incorrect by a factor of 2. However, at 60 nM and on the basis of the k_2_/K and k_−2_ values, the k_i_ value would be 0.65 × 10^−5^ s^−1^, i.e., it would take 28 h to reach 50% of the equilibrium value (25% inactivation). The global equilibrium constant is thus irrelevant and the only important factor is the k_2_/K second-order rate constant. Note that the global equilibrium constant and k_−2_ values are also in fair agreement with those of Lahiri et al. [17].

### 3.2. OXA-163

#### 3.2.1. Interaction of OXA-163 with Ceftazidime

The value of A_260_ was monitored for a period of 2 min unless otherwise stated. The final enzyme concentration was 115 nM in 500 µL of substrate.

At ceftazidime concentrations of 40 and 80 µM, the hydrolysis was first-order. The K_m_ value can thus be considered as much larger than 80 µM and this substrate can easily be used as a reporter substrate at this concentration without any correction for protection. The large K_m_ value is in agreement with the data of Poirel et al. (K_m_ > 2 mM [8]). However, we find a k_cat_/K_m_ value of 57 000 ± 5000 M^−1^s^−1^ which is significantly lower than the reported value (10^6^M^−1^s^−1^). This is why we performed a titration of the enzyme with avibactam (see below). Note however that the published k_cat_/K_m_ value calculated on the basis of a k_cat_ of 200 s^−1^ and a K_m_ > 2 mM [8] would be < 100,000 M^−1^s^−1^.

#### 3.2.2. Titration by Avibactam

Since our k_cat_/K_m_ value was significantly lower than the published one [8], we performed a titration of the enzyme by avibactam. In a total volume of 40 µL, 0.088 nmol of enzyme were incubated at 22 °C with 0, 0.025, 0.05, 0.075 and 0.10 nmol of avibactam. After 70 min, the residual activity was estimated by the addition of 480 µL of ceftazidime. The results, shown in Table A1 indicate a good linear fit that shows that total inactivation occurs at an [Avi]/[enzyme] ratio of about 1.12. The enzyme is thus pure and fully active and this confirms our k_cat_/K_m_ value.

#### 3.2.3. Inactivation of OXA-163 by Avibactam in the Presence of 80 µM Ceftazidime

The enzyme concentrations ranged from 133 to 266 nM and those of avibactam from 4.8 to 24 µM. The measured rate constants (k_i_) ranged from 1 to 4.8 × 10^−2^ s^−1^. A linear plot of k_i_ vs. [Avi] yields a k_2_/K value of 1720 ± 75 M^−1^s^−1^ (Figure 1). Any K value would be significantly higher than 25 µM.

#### 3.2.4. Reactivation Rate

With 19.2 and 24 µM avibactam (k_i_ = 0.03 and 0.04 s^−1^), less than 0.5% of activity was detected after 120 s, indicating a k_−2_ value lower than 1.5 × 10^−4^ s^−1^.

A preliminary experiment indicated that activity recovery was very slow and that the half-life of the adduct was likely to be > 24 h. The enzyme (1 µM, 100 µL) was incubated with 2 µM avibactam. After 20 min, the solution was diluted 1000-fold (residual avibactam concentration: 2 nM) and the activity determined on 450 µL aliquots by the addition of 50 µL of 400 µM nitrocefin. A control sample without avibactam was similarly treated. Activity was determined after 0, 21 (recovery: 28%) and 45 (recovery: 41%) h. To calculate the k_−2_ value, it must be considered that the measured rate constant = k_−2_ + k_f_ and that the recovery is limited by the global equilibrium constant as above with OXA-24.

Successive approximations indicate that K_eq_ is close to 3 nM. On this basis, the activity at equilibrium is expected to be 60% of that of the control (note that we did not observe any decrease of the activity of the control over the 45 h period) and the reactivation rate constant is 8.3 × 10^−6^ s^−1^. The calculated k_f_ value at 2 nM avibactam is 3.4 × 10^−6^ s^−1^ so that k_r_ is 4.9 × 10^−6^ s^−1^ (half-life: 39 h). The global equilibrium constant is thus 4.9 × 10^−6^ M/1700 = 3 nM. It should be noted that the calculated k_f_ value is probably somewhat too large since a 2 nM avibactam concentration is no longer much larger than that of the enzyme but this does not significantly influence the conclusions.

#### 3.2.5. Global Equilibrium Constant

A value of 3 nM is given above, but due to the very low value of k_−2_, it is careful to consider that this value might be incorrect by a factor of 2. However, at 3 nM and on the basis of the k_2_/K and k_−2_ values, the k_i_ value would be about 1 × 10^−5^ s^−1^, i.e., it would take 19 h to reach 50% of the equilibrium value (25% inactivation). It would thus be difficult to directly determine the global equilibrium constant.

### 3.3. OXA-427

#### 3.3.1. Hydrolysis of Ceftazidime and Cephalothin

##### Ceftazidime

Both in the presence and absence of NaHCO_3_, the enzyme activity exhibited a broad maximum between pH 6 and pH 9. Experiments were performed at pH 6.2 in the absence of NaHCO_3_ and 6.7 in its presence (100 mM sodium phosphate).

The hydrolysis of 80 µM ceftazidime by 32 nM enzyme was monitored during 5 min and the (quasi-linear) rate determined over the first 30s. The presence of 40 mM NaHCO_3_ increased this initial rate by about 25%. By contrast between 270 and 300 s, the presence of NaHCO_3_ increased the rate by a factor of 2.8. Thus, with ceftazidime, substrate-induced inactivation seems to take place in the absence of CO_2_ as it does with most (if not all) OXA enzymes. The kinetic parameters were thus determined by initial rate measurements only in the absence of NaHCO_3_ and both by initial rate and complete time-course measurements in its presence. Under these conditions, a K_m_ value of 96 ± 13 µM (k_cat_ = 8.1 ± 1.1 s^−1^) was obtained and there was no significant difference between 10 and 40 mM NaHCO_3_. In the absence of NaHCO_3_, initial rate measurements (performed over 1 min or less) yielded a K_m_ value of 68 ± 7 µM with k_cat_ = 5 ± 0.3 s^−1^.

##### Cephalothin

With this substrate, the presence of NaHCO_3_ did not result in significant differences in the hydrolysis rates measured as above between 0 and 30 s or after 270 s. The K_m_ value was found to be very low (2 µM) so that the kinetic parameters were determined on the basis of complete time-course measurements with an initial cephalothin concentration of 20 µM. The kinetic parameters were k_cat_ = 0.47 ± 0.1 s^−1^ and K_m_ = 1.7 ± 0.4 µM in the absence of NaHCO_3_ and 0.43 ± 0.034 s^−1^ and 2.0 ± 0.2 µM in its presence. Thus, the presence of NaHCO_3_ does not seem to influence the kinetic parameters, at least over an 80–100 s time-course.

#### 3.3.2. Inactivation by Avibactam with Ceftazidime as Reporter Substrate

##### In the Absence of NaHCO_3_

The enzyme (0.16–0.32 nM) was incubated with 83 µM ceftazidime and 23–54 µM avibactam (total volume, 420–440 µL). The hydrolysis of ceftazidime was monitored over a period of 180 s. It was verified that the activity was close to zero (<2%) after 10 min. The pseudo-first-order inactivation rate constants (k_i_) varied from 0.8 to 2.13 × 10^−3^ s^−1^. A linear regression of k_i_ vs. [Avi] yielded a value of 360 ± 14 M^−1^s^−1^ for (k_2_/K)_obs_. As shown in Figure 2, there was no indication of a deviation from linearity at the highest avibactam concentration.

After correction for the protection by ceftazidime (with K_m_ = 68 µM) a final value of 800 ± 100 M^−1^s^−1^ was found for (k_2_/K)_corr_.

##### In the Presence of 40 mM NaHCO_3_

The experiments were performed under the same conditions and the avibactam concentrations ranged from 20 to 56 µM. A (k_2_/K)_obs_ value of 470 ± 70 M^−1^s^−1^ was found, yielding a corrected value of 870 ± 200 M^−1^s^−1^ after taking the K_m_ value (96 µM) into account.

#### 3.3.3. With Cephalothin as a Reporter Substrate

Hydrolysis time-courses of 20 µM cephalothin were recorded in the presence of 88 and 170 µM avibactam. The k_i_ values were proportional to the inactivator concentrations, yielding k_2_/K values of 70 ± 10 M^−1^s^−1^ in the absence of bicarbonate and of 86 ± 11 M^−1^s^−1^ in its presence (40 mM). The large errors are due to the very small ΔA values. The presence of bicarbonate does not seem to result in a significant difference and an average value of 80 ± 20 M^−1^s^−1^ was obtained. After correction for the K_m_ value of cephalothin (2 µM, see above) a (k_2_/K)_corr_ value of 880 ± 200 M^−1^s^−1^ was obtained, in excellent agreement with the values reported above with ceftazidime as a reporter substrate. Note that these results also indicate that the K value should be significantly larger than 170 µM.

#### 3.3.4. Reactivation

The enzyme (15 µM) was incubated for 30 min with 23 µM avibactam in a total volume of 45 µL of 50 mM NH_4_HCO_3_ adjusted to pH 6.8 with acetic acid. The residual activity was below 2%. The solution was then diluted 100-fold with 100 mM sodium phosphate buffer containing 50 mM NaHCO_3_ (final pH: 6.8). After 22 h, the recovered activity was not higher than 16 ± 8% of that of the control (note that the control itself lost 30% of its activity). This yields a maximum global equilibrium constant (K_eq_) value of 15 ± 7 nM and since k_−2_ = KK_eq_/k_2_, the maximum value of k_−2_ is 1.3 ± 0.7 × 10^−5^ s^−1^ (half-life > 14 ± 7 h). The large errors are due to the low activities that were measured.

## 4. Mass Spectrometry

### 4.1. OXA-427

Free OXA-427 (E) yielded a mass of 28,213 (expected: 28,212.9). After reaction with an excess of avibactam, two peaks were detected, one at 28,478 (E + avibactam) and one at 28,399 (E + avibactam − SO_3_). When a stoichiometric amount of avibactam was used, the free enzyme peak completely disappeared, indicating that the enzyme was pure. Another experiment was performed in which about 60% of the enzyme was inactivated by using a sub-stoichiometric quantity of avibactam. Three peaks are clearly visible (Figure 3). Moreover, the 28,399/28,478 ratio did not change when the contact time was increased from 30 min to 24 h.

### 4.2. OXA-163

Similar experiments were performed with OXA-163 (free enzyme: 28,502, expected 28,502.2). After complete inactivation, two peaks (28,767 and 28,687) were obtained and the peak corresponding to the free enzyme completely disappeared. The enzyme was also partially (about 30%) inactivated by a substoichiometric amount of avibactam. After 30 min of contact, three peaks were detected (Figure 4): 28,502 (free enzyme), 28,767 (E + avibactam) and 28,687 (E + avibactam − SO_3_). Again, the 28,687/28,767 ratio was rather high and was not modified by increasing the contact time to 22 h.

These results seem to indicate that, with both enzymes, the loss of SO_3_ is concomitant with the acylation reaction. The proportion of adduct showing a loss of SO_3_ is much higher than those observed by others with other enzymes [16,20,21,22].

## 5. Discussion

There are very few studies about the interactions between avibactam and class D β-lactamases. Reliable kinetic parameters can only be found in the papers by Ehmann et al. [16] and Lahiri et al. [17]. Generally, only MIC values for OXA-producing strains are compared in the presence and absence of this β-lactamase inactivator (as e.g., [23,24]). The available data indicate that OXA-10 [16] and OXA-24 [17] are not very susceptible to avibactam with k_2_/K values of 11 and 51 M^−1^s^−1^, respectively. OXA-48 [16] and OXA-23 [17] are significantly more susceptible with k_2_/K values of 1400 and 300 M^−1^s^−1^ respectively. However, avibactam is often administered in combination with ceftazidime and, to our knowledge, no analyses of the interactions between avibactam and ceftazidime-hydrolyzing class D β-lactamases have been published. In this paper, we devoted our attention to OXA-163 and OXA-427 that both exhibit significant activities on ceftazidime. Table 1 summarizes the kinetic parameters that characterize the interactions between some OXA enzymes and avibactam and ceftazidime. We first determined the exact kinetic parameters of both enzymes with this third-generation cephalosporin. With OXA-163, K_m_ was >> 80 µM and k_cat_/K_m_ was 57,000 M^−1^s^−1^ indicating that k_cat_ was >> 5 s^−1^. With OXA-427, in the presence of NaHCO_3_, K_m_ was 96 µM and k_cat_ 8 s^−1^. In the absence of NaHCO_3_, the K_m_ and k_cat_ values, when measured over a 30–60 s time scale, were both somewhat lower but not significantly different. Interestingly, if one assumes a ceftazidime periplasmic concentration of 10–20 µM, the rates of hydrolysis by both enzymes would be similar (0.5 s^−1^ × E0, where E0 is the periplasmic enzyme concentration). For both these enzymes, the second-order acylation rate constants by avibactam (OXA-163: 1700 M^−1^s^−1^, OXA-427: 800 M^−1^s^−1^) were significantly larger than those observed for OXA-10 and OXA-24. The deacylation rates were extremely low, resulting in covalent adducts exhibiting very long half-lives, so that the k_−2_ parameter was irrelevant and the only important constant was k_2_/K. This seems to be a general property of class D enzymes. The time necessary to inactivate the enzymes that exhibit very low k_2_/K values (e.g., OXA-10 and OXA-24), at avibactam concentrations of 10–20 µM would probably be too long to be of physiological significance (see Table 1). However, for the enzymes that do not hydrolyze ceftazidime, this point is not very relevant.

With OXA-427, substrate-induced inactivation (i.e., decarboxylation of the active-site Lys residue) did not seem to significantly influence the kinetic parameters of ceftazidime and cephalothin when measured over short periods of time. Similarly, the absence of bicarbonate in the reaction mixture did not influence the rate of inactivation by avibactam.

The mass spectrometry results showed that the purified proteins exhibited the expected masses, that they were pure and that all the enzyme molecules were active. Another interesting observation was that the covalent adducts exhibited a loss of SO_3_ that was significantly larger than with other enzymes with which this loss was rather low [16,20,21,22]. Although protein mass spectrometry is not very good as far as the quantitative aspects are concerned, the ratio [adduct]/[adduct − SO_3_] did not seem to increase with time, indicating that the loss of SO_3_ probably occurs during the acylation reaction. Due to the very high stability of the adducts, this observation, although interesting from a mechanistic point of view, is not very relevant when one considers the practical problem of inactivating OXA-type β-lactamases. Note that the mass spectrometry results also confirm the high stability of the covalent adducts.

OXA-427 deserves an additional comment. We have observed (unpublished results) that the carbapenems often exhibit very low K_m_ values with this enzyme. In consequence, the presence of carbapenems would significantly slow down the inactivation process. This might be worth remembering if a combined avibactam/carbapenem therapy was considered in the future.

## 6. Conclusion

We have shown that two ceftazidime-hydrolyzing OXA enzymes (OXA-163 and OXA-427) were more susceptible to avibactam than several other class D β-lactamases, with the exceptions of OXA-48 and, possibly, OXA-23. This observation is of special clinical relevance since avibactam is often used in combination with ceftazidime. When a class D enzyme is identified as an ESBL, the determination of its susceptibility to avibactam would thus be of major interest. This can easily be done with the help of the reporter substrate method utilized in the present contribution. It does not even require the complete purification of the OXA enzyme but one should make sure that the latter is the only β-lactamase in the analyzed sample.

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
