# Peer review of "Interactions between Avibactam and Ceftazidime-Hydrolyzing Class D β-Lactamases"

_biomolecules, 2020, doi:10.3390/biom10030483_

Round 1

Reviewer 1 Report

In the manuscript titled “Interactions between avibactam and ceftazidime-hydrolysing class D β-lactamases.”, the authors described a detailed kinetic study of a CHDL inhibitor, avibactam, and two beta-lactam based antibiotics, ceftazidime and cephalothin against three OXA beta-lactamases, OXA-24, OXA-163, and OXA-427.  They measured and reported key chemical kinetics rate constants for catalytic hydrolysis reaction of the target enzymes against the selected antibiotics under various conditions. This is an informative kinetic study for class D beta-lactamases, and will benefit the antibiotics related research communities. There are several key issues need to be address before its publication.

1) Section 2.3 Kinetic measurements is very poorly written. Key kinetics rate constants are not labeled. Key equations are not presented clearly. It is extremely hard to follow this part, which is the key to understand the kinetics in this study.

2) Mass spectroscopy study does not seem to be integrated well in this study. It is not clear what has been learnt through the mass spectroscopy study of the enzyme/ligand complexes. More discussion could help to clarify this.

3) It is puzzling that kinetic data related to cephalothin are not presented in Table 1, which contains the key discovery of this study.

Author Response

We thank the referees for their careful reading of the paper.

It should first be noted that the referees did not receive the version that we submitted. For reasons that are not clear to us, the following happened :

  • The constants disappeared from the reaction scheme between lines 123 and 124 (now). This explains why the results were « not clearly presented » and that « section 2.3, kinetic measurements is poorly written » (referee 1) and also explains referee’s 2 point 2.
  • The subscripts and the superscripts disappeared as well as the italics (referee’s 2 points 5 and 10).

Other points.

Referee 1.

  • The introduction has been completely rewritten. However, a more complete review of the properties of class D beta-lactamases would be much too long.
  • Point 1 : see above.
  • Point 2 : we added some comments in the discussion about the mass spectra. These data show that the enzymes are pure and confirm the high stability of the covalent adduct. Also, the high (M + 185)/(M + 265) ratio is interesting and specific to these enzymes.
  • Point 3 : we think there is some misunderstanding : the key point of the paper is that two enzymes that hydrolyze ceftazidime are sensitive to avibactam. The kinetic data concerning this substrate are mentioned. Cephalothin was only used as a reporter substrate in the study of OXA-427. We can add the kinetic parameters in the legend but it would not be very useful.

Reviewer 2 Report

The manuscript by J-M Frere et al focuses on an important aspect of Class D beta-lactamases, that of diversity in their interaction with b-lactam substrates and inactivators. The authors have undertaken an in-depth characterization of the enzymatic activity of two of OXA-24  derivatives, OXA-427 and OXA-163.

Their finding that Class D beta-lactamases (such as OXA-427 and OXA-136) that hydrolyze ceftazidime are more susceptible to inactivators such as avibactam is quite interesting; it suggests that along with characterization of the substrate spectrum of this enzymes, the inactivator spectrum needs also to be characterized.

The manuscript is technically sound and overall well written. Few minor issues are described below.

  1. Line 46 “catanevertheless”
  2. It will be very useful to non-experts to identify the microscopic kinetic parameters on the kinetic equation (between lines 117 and 118); which steps in these kinetic equation are characterized by k1, k-1 and so on, and what is kr and kf?
  3. In equation 3, please use the multiplication symbol to avoid misreading of this equation.
  4. Line 113, I am assuming the enzyme activity is “poor” at low enzyme enzyme concentrations (obtained after dilutions), If so, it has to be made clear.
  5. Please keep the names of bacterial species in italic, line 151.
  6. Line 201, what is (106)? What are the units? I am assuming it is the value of kcat/Km determined by Poirel et al. Please provide the units.
  7. Lie 215, extra dots after (Figure 1).
  8. Line 270, the amount of the enzyme in the assay is good to know, but the concentration would be better to be able to reproduce these data by others.
  9. Figure 3, The “%”symbol on the y-axis is very small, please increase the font size. Also, “mass” as the unit of the X-axis is not accurate, this is “Z/e”; unless the authors mentioned that these charged species are singly charged.
  10. An overall comment, please use the correct format for the kinetic parameters: italic k (K), and subscript number. It is likely that the fromatting disappeared upon manuscript formatting.

Author Response

We thank the referees for their careful reading of the paper.

It should first be noted that the referees did not receive the version that we submitted. For reasons that are not clear to us, the following happened :

  • The constants disappeared from the reaction scheme between lines 123 and 124 (now). This explains why the results were « not clearly presented » and that « section 2.3, kinetic measurements, is poorly written » (referee 1) and also explains referee’s 2 point 2.
  • The subscripts and the superscripts disappeared as well as the italics (referee’s 2 points 5 and 10).

Other points.

Referee 2.

  • Point 1 : done
  • Point 2 : see above
  • Point 3 : done
  • Point 4 : the text has been modified to clarify this point
  • Point 5 : see above
  • Point 6 : the units are M-1s-1 (added as requested). It should read 106 M-1s-1 (see above)
  • Point 7 : done.
  • Point 8 : done.
  • Point 9 : the size of the % symbol has been increased. Concerning the abscissa, I am afraid we have to disagree with the referee. According to our mass spec specialist, « mass » is correct because it is a deconvoluted spectrum. Note that you find the same « mass » abscissa in the mass spectra in references 16 and 20-22. We have added the units (amu for atomic mass units) between parentheses.
  • Point 10 : see above.

Round 2

Reviewer 1 Report

The authors have addressed the raised concerns. The manuscript could be accepted for publication.